# Can the Firmness, Weight, and Size of Blueberry Fruit Be Enhanced through the Application of Low Amounts of Calcium to the Soil?

**DOI:** 10.3390/plants13010001

**Published:** 2023-12-19

**Authors:** Juan Hirzel

**Affiliations:** Quilamapu Regional Research Center, Instituto de Investigaciones Agropecuarias, Avenida Vicente Méndez 515, Chillán 3780000, Chile; jhirzel@inia.cl

**Keywords:** *Vaccinium corymbosum*, calcium fertilization, fruit, firmness, size, weight

## Abstract

Background: The firmness, weight, and size of blueberries are vital for commercial success of this crop. Fertilization is a key agronomic management practice that affects fruit quality, where calcium (Ca) plays a critical role. This study aimed to assess the impact of low levels of soil-dosed Ca in carboxylic acid form on fruit size, weight, firmness, and residual soil fertility. Methods: The study focused on two varieties of blueberries, Duke and Legacy, over two consecutive growing seasons on three commercial farms located in south-central Chile. This study consisted of five treatments, ranging from 0 to 4.0 kg Ca per hectare. Results: The highest firmness values observed for Duke were between 164 and 186 g mm^−1^, with size values ranging from 15.7 to 16.9 mm, and weight observations ranging from 1.60 to 1.76 g. On the other hand, Legacy showed firmness values between 163 and 173 g mm^−1^, with size values ranging from 16.2 to 17.2 mm, and weight observations ranging from 2.01 to 2.40 g. Conclusion: The application of low Ca rates to the soil did not impact the size, weight, or firmness of ‘Duke’ and ‘Legacy’ blueberries. There was a positive correlation between the Ca soil application and the concentration of exchangeable Ca.

## 1. Introduction

Human consumption of nutritious foods such as blueberries is gaining importance. The primary blueberry-producing countries include the United States, Canada, Chile, Perú, and Spain, which collectively produced over 845,000 t in 2019 [1]. To improve crop profitability, agronomic management practices such as pruning and fertilization play a critical role [2,3,4]. Such practices optimize fruit-quality attributes, especially firmness, size, weight, and total soluble solids [3,5,6,7]. These attributes show quantitative differences in value among cultivars [3,8,9], seasons, and production zones [3,10,11,12,13], which makes it difficult to cite standard reference values.

Calcium (Ca) is a nutrient used to improve fruit quality. It is commonly applied pre-harvest to extend postharvest shelf life [14,15,16]. Calcium functions include structural roles in cell walls, membrane stability, as well as chemical messenger communication between different plant organs and tissues [17]. Increasing calcium concentration in the fruit has a beneficial effect on fruit firmness because Ca-pectin interactions can regulate control pectin depolymerization and hydrolysis, thereby increasing postharvest fruit shelf life [18]. Olmedo et al. [19] have reported that the calcium content associated with cell wall pectin polysaccharides affects the maximum compressive strength (hardness) of ‘Emerald’ (firm cv.) and ‘Jewel’ (softer cv.) blueberries during postharvest storage. The authors have suggested that this relationship could be explained by the effect of calcium on the binding of unesterified pectin and the consequent reduction in cell wall degradation. Calcium uptake is mainly regulated by maintaining a concentration gradient in the roots; it is then quickly distributed to other plant organs or stored in the vacuoles of root cells, thus maintaining a low concentration in the cytoplasm [20].

There is little published information regarding the optimum concentration or critical range of exchangeable Ca in soil to maximize blueberry crop yield. Komosa et al. [21] reported a critical concentration range of 0.5 to 1.5 cmol_+_ kg^−1^ for mineral soils, while Pinochet et al. [22] identified a concentration of 0.6 cmol_+_ kg^−1^ at a soil depth of 0–20 cm as the appropriate or critical level of exchangeable Ca for blueberry cultivation in the volcanic soils of southern Chile.

Regarding calcium soil application, Angeletti et al. [23] reported that applying calcium sulfate (0.06 kg m^−2^) (52.8 kg Ca applied in 4000 m^2^ cropped on one ha) increased calcium content in ‘O’Neal’ and ‘Bluecrop’ blueberry fruits, while reducing postharvest firmness and weight loss (after 23 days of storage at 2 °C) compared to controls without calcium application. Additionally, postharvest respiration was lower in treatments with calcium soil application. Garvarino [24] reported an increase in fruit firmness in blueberry cv. Ochocklonne (*Vaccinium virgatum* L.), as the doses of Ca (1 and 2 L ha^−1^), complexed with carboxylic acid (Calcium Sprint), applied during the flowering to fruit set period.

Davis and Strik [25] conducted a field experiment on the response of blueberry quality and nutritional characteristics to soil Ca application. They used ‘Elliott’ blueberry and observed that the application of sawdust as a mulch (141 m^3^ ha^−1^) increased the Ca concentration in soil, leaves, and fruit compared to the control, thus increasing the soil pH; however, fruit firmness was not affected. It is worth mentioning that sawdust presented a Ca concentration of 815 mg kg^−1^ [26]. In some field experiments with foliar application of Ca in blueberries, no response has been found for increasing fruit Ca concentration or quality attributes such as firmness in ‘O’Neal’ [14], ‘Draper’ and ‘Legacy’ [16], ‘Alapaha’ and ‘Powderblue’ rabbiteye (*Vaccinium virgatum* Aiton) [27], or fruit weight in ‘Draper’ and ‘Bluecrop’ [28]. However, soil Ca concentration was high (23.4 cmol_+_ kg^−1^) in the Manzi and Lado [14] experiment, whereas Vance et al. [16], Smith [27], and Arrington et al. [28] did not report soil Ca concentration. In contrast, a field experiment conducted in Poland showed that some of the evaluated products increased fruit firmness and fruit weight when foliar Ca was applied to ‘Bluecrop’ blueberries at a soil Ca concentration of 4.76 cmol_+_ kg^−1^ [29]. Similarly, another field experiment on ‘Liberty’ blueberries conducted by Lobos et al. [30] found a positive effect on both fruit firmness and fruit weight when foliar Ca was applied; however, soil Ca concentration was not mentioned. Gerbrandt et al. [31] showed in two seasons and three experimental sites that foliar application of Ca with Ca silicate or Ca chloride formulations at the petal drop or green fruit stage was able to reduce early fruit drop and increase fruit weight in cv. ‘Draper’; however, these authors did not mention the chemical properties of the soils in which these experiments were conducted. The above information suggests that the response to soil or foliar application of Ca may depend on the Ca concentration in the soil and the cultivar being evaluated.

Regarding reference values for some quality attributes of blueberry fruit, in a 6-year experiment conducted at Oregon State University North Willamette Research and Extension Center, Strik et al. [3] reported mean values of 2.19 and 2.12 g for ‘Duke’ and ‘Legacy’ fruit weight, respectively. During a 4-year evaluation, mean fruit firmness values determined with the Firmtech equipment in the same experiment were 177 g mm^−1^ for ‘Duke’ and 170 g mm^−1^ for ‘Legacy’. Firmness was affected by evaluation year and cultivar and the interaction between the two factors, whereas fruit weight was not affected by evaluation year but was affected by cultivar and the year × cultivar interaction.

The main blueberry cultivars grown in Chile are ‘Legacy’ (3217 ha, 18.4%) and ‘Duke’ (2524 ha, 14.4%) [32,33] whose quality attributes have been affected by factors such as increased temperature during the summer. It is worth noting that the application of Ca to the soil in the blueberry crop is a common practice that can improve fruit quality attributes, and that the response can vary among edaphic conditions, climate, and application techniques. Considering that the first phase of fruit growth is characterized by rapid cell division [34], and that this process is promoted by the presence of calcium and cytokinins [35], the application of low doses of calcium with sources that facilitate its availability in early stages of flower and fruit development, associated with low natural availability when soil temperature is low, could contribute to improve fruit firmness. The hypothesis of our work is that the application of Ca to the soil at lower rates through the formulation of carboxylic acids between flowering and 14 days after flowering can improve some quality attributes in fruit and also affect some chemical properties of the soil. The objective of the present study was to evaluate the effect of low doses of Ca applied to the soil as a carboxylic acid formulation between flowering and 14 days after flowering on the firmness, size, and weight of ‘Duke’ and ‘Legacy’ blueberry fruits, and on residual soil fertility under commercial growing conditions in south-central Chile.

## 2. Materials and Methods

### 2.1. Seasons and Experimental Sites

The present study was conducted at three commercial blueberry farms located in south-central Chile, with Xerorthends (Entisol), Xerochreps (Inceptisol), and Melanoxerands (Andisol) soils [36] during the 2020–2021 and 2021–2022 seasons. The climate at the sites is temperate Mediterranean characterized by a hot, dry summer and a cold, wet winter. Annual precipitation was 576 and 649 mm for the 2020–2021 and 2021–2022 seasons, respectively, which was concentrated from late fall to early spring. The average temperature was 14.3 and 13.5 °C, while evaporation was 1060 and 940 mm for the 2020–2021 and 2021–2022 seasons, respectively [37]. The fields were located in Santa Cruz de Cuca for Entisol (36°39′44″ S; 72°26′22″ W), Larqui for Inceptisol (36°44′34″ S; 72°12′51″ W), and Capilla for Andisol (36°32′08″ S; 71°54′59″ W). Soil physicochemical properties at 0–30 cm depth are shown in Table 1.

### 2.2. Initial Soil Analysis

Composite samples were collected manually from the topsoil layer (0–30 cm) at the beginning of the experiment. All samples were air dried and sieved (2 mm mesh). Soil pH was determined in 1:2.5 soil:water extracts. Soil organic matter (OM) was measured by the Walkley–Black wet digestion method [38]. Soil available N (NO_3_-N and NH_4_-N) was extracted with 2 M KCl solution and calculated by colorimetry using a segmented flow spectrophotometer (autoanalyzer, Skalar Analytical BV, Breda, The Netherlands). Soil extractable phosphorus (P) was extracted with 0.5 M NaHCO_3_ (Olsen P) and determined by the molybdate–ascorbic acid method. Exchangeable Ca, magnesium (Mg), potassium (K), and sodium (Na) were determined by a 1 M NH_4_OAc extraction followed by flame spectroscopy, absorption (Ca and Mg) and emission (K and Na). Soil exchangeable aluminum (Al) concentration was measured with a 1 M KCl extraction by absorption spectroscopy, while sulfur (S as SO_4_^2−^-S) was determined with 0.01 M calcium phosphate and by turbidimetry. Soil iron (Fe), manganese (Mn), zinc (Zn), and copper (Cu) concentrations were determined in diethylenetriaminepentaacetic acid (DTPA) extract by atomic absorption spectrometry [39]. Boron (B) was measured by colorimetry in a hot water solution. Soil texture was analyzed by the Bouyoucos hydrometer method.

### 2.3. Crop Management

‘Duke’ (early harvest) and ‘Legacy’ (mid-season harvest) blueberries (*Vaccinium corymbosum* L.) were grown on the three soils. The age of the orchard ranged from 6 to 10 years (orchards at the peak of production with yields per plant between 10 and 15 t ha^−1^ for ‘Duke’ and 15 and 20 t ha^−1^ for ‘Legacy’). The planting distance of the three fields was 3 m between rows and 1 m above rows (3333 plants ha^−1^). Fertilizer rates applied were 80 kg N ha^−1^ (ammonium sulfate), 60 kg P_2_O_5_ ha^−1^ (monoammonium phosphate), 120 kg K_2_O ha^−1^ (potassium sulfate), and 30 kg MgO ha^−1^ (magnesium sulfate) by fertigation for the three soils in both seasons and for both blueberry cultivars during the growing season. In addition, boron was applied by fertigation in Entisol at a rate of 2 kg ha^−1^ yr^−1^. Irrigation consisted of water replenishment in the evaporation pan adjusted by the crop coefficient (K_c_). The phytosanitary management used by the growers was similar among orchards and cultivars.

### 2.4. Treatments

The treatments evaluated were five doses of Ca fertilization applied to the soil and dissolved in water (simulating fertigation application); the doses were 0, 0.5, 1.0, 2.0, and 4.0 kg Ca ha^−1^, which were totally applied between start of flowering and 14 days after flowering. The fertilizer used was Calcio Sprint (5% Ca), with commercial doses corresponding to 0, 10, 20, 40, and 80 L ha^−1^ for each treatment.

### 2.5. Fruit Sample Collection and Analysis

Fruit samples were collected at the commercial harvest stage (fruit with 100% blue color) during the first and second weeks of harvest in each season to determine the quality attributes of fruit firmness, size, and weight. Fruit sampled was 100% for ‘Duke’ and 80% for ‘Legacy’. The remaining 20% of the ‘Legacy’ fruit was mechanically harvested due to inferior quality during the third week of harvest. Fruit was harvested between 8:30 and 10:00 a.m. in plastic trays and transferred to a thermal insulation structure (Igloo 144 L, Igloo Products Corp, TX, USA). They were transported to the Fruit Analysis Laboratory of the Instituto de Investigaciones Agropecuarias (Chilean Agricultural Research Institute), Quilamapu Regional Research Center in Chillán, Chile (36°35′43″ S; 72°05′16″ W) for immediate determination of fruit firmness, size, and weight. Firmness and size of 60 fruits from each sample were measured individually with a FirmPro instrument (HappyVolt, Santiago, Chile), and fruit weight was determined with a digital balance (model 100A-300M, Precisa, Dietikon, Switzerland).

### 2.6. Soil Sample Collection and Analysis

Soil samples were collected at the end of the second season (April 2022) at 0–30 cm depth in each experimental unit. Sampling was performed in the root zone (fertilized zone) with 10 controls per experimental unit. Water pH, electrical conductivity (EC), and exchangeable Ca were analyzed in each sample using the methods described above [38]. EC was determined in 1:5 soil:water extracts. For each sample, a saturated liquid extract was prepared by vacuum filtration in which water pH, EC, and available Ca were determined using the methods mentioned above for initial soil analysis [38].

### 2.7. Experimental Design and Statistical Analysis

The experimental design for each blueberry cultivar and harvest week was a completely randomized block design with a split-split plot arrangement, and for the soil analysis the experimental design was a randomized block with a split-plot arrangement. For fruit analysis, the main plots were the two seasons, the split-plots were the three soils, and the split-split plots were the five Ca rates with five replications (n = 150). For soil analysis the main plots were the three soils, and the split plots were the five Ca rates with five replications (n = 75). Results were analyzed by ANOVA and Tukey’s test (*p* = 0.05) using the SAS PROC MIXED Model procedure [40]. In the case of significant interactions, contrast analysis was used to compare the effects of treatments separately.

## 3. Results

Table 2 shows the statistical analysis of ‘Duke’ fruit in the first and second weeks of harvest. In the first week, the three quality attributes evaluated were not affected by either Ca rate or interactions between Ca rate and other sources of variation. Firmness was affected by soil and the season × soil interaction, while both fruit size and fruit weight were affected by season, soil, and the season × soil interaction. In the second week, the Ca rate affected both fruit size and fruit weight, but there was no interaction with other sources of variation. Firmness was affected by season, soil, and season × soil interaction, while fruit size and fruit weight were affected by season and soil, although only fruit size showed a season × soil interaction.

The season × soil interaction for the first week of harvest for ‘Duke’ indicated that the highest values of fruit quality attributes in the first season occurred on Inceptisol. Fruit firmness, fruit size, and fruit weight were 16%, 7.5%, and 17% higher, respectively, than the mean values for the other soils (Table 3). The highest value of fruit firmness for ‘Duke’ in the second season also occurred in the Inceptisol; however, both fruit size and fruit weight were similar in the Inceptisol and Andisol. When comparing between seasons, the mean firmness in the three soils was similar (166 and 165 g mm^−1^, respectively), while the mean in the three soils for fruit size and fruit weight was higher in the second season (Table 3).

The season × soil interaction for the second week of harvest for ‘Duke’ indicated that the highest values for both firmness and fruit size in the first season occurred in the Inceptisol (see Table 4). Firmness in the Inceptisol was 15.5% and 32.9% higher than in the Andisol and Entisol, respectively, while fruit size values in the Inceptisol were 5.0% and 19.9% higher than in the Andisol and Entisol, respectively (Table 4). Firmness in the Inceptisol in the second season was 9.3% and 14.7% higher than in the Andisol and Entisol, respectively. Fruit size was similar in the Inceptisol and Andisol with a mean value 15% higher than in the Entisol (Table 4). When comparing the two seasons, a 6% and 4% decrease in both fruit firmness and fruit size, respectively, was observed compared to the results of the first season (Table 4).

Fruit weight in the second week of harvest was 5% higher in the first season. On average, this value was 39% higher in the Inceptisol and Andisol than in the Entisol (Table 5). No clear effect of Ca dose on fruit weight was observed, as the doses of 0.5 and 2 kg Ca ha^−1^ were significantly similar to the control without Ca application (Table 5). However, in quantitative terms, the mean of the four Ca doses increased fruit weight by 9% compared to the control.

Statistical analysis for ‘Legacy’ and the first and second weeks of harvest showed that all three quality attributes evaluated were influenced by season, soil, and the season × soil interaction (Table 6). Ca rate as an independent factor did not affect fruit quality attributes at either harvest week; however, both fruit size and fruit weight at the second harvest week were affected by the season × Ca rate interaction (Table 6).

The season × soil interaction for the first week of harvest for ‘Legacy’ in the first season indicated that the highest fruit firmness occurred in Entisol and Andisol, which were on average 12.7% higher than those in Inceptisol (Table 7). Fruit size in Inceptisol was 2.5% higher than in the other two soils. Finally, fruit weight was 13.6% higher in Inceptisol than in Andisol, and there was no significant difference with Entisol (Table 7). In addition, there was an inversely proportional relationship between fruit firmness and weight (R = −0.5) and a directly proportional relationship between fruit size and weight (R = 0.6) in the first week of harvest in the first season for ‘Legacy’. Fruit firmness for the second season in the Entisol was 5% and 20% greater than in the Andisol and Inceptisol, respectively (Table 7). Fruit size in the Inceptisol was 7.8% greater than the combined means of the other two soils. Finally, fruit weight in the Inceptisol was 17% and 29% higher in the Inceptisol than in the Entisol and Andisol, respectively (Table 7). In addition, there was a directly proportional relationship between fruit size and fruit weight (R = 0.99) during the first week of harvest in the second season for ‘Legacy’. When comparing both seasons, a 4.7% decrease in fruit firmness was observed, while both fruit size and fruit weight increased by 3.1% and 12.9%, respectively, compared to the first season (Table 7).

The season × soil interaction in the second week of harvest for ‘Legacy’ showed that the highest fruit firmness in the first season was in the Entisol and Andisol, which were on average 11.2% higher than in the Inceptisol (Table 8). However, the highest value for both fruit size and fruit weight was recorded in the Inceptisol, where fruit size was 11.5% higher than the combined means of the other two soils, while fruit weight was 21% and 42% higher than in the Andisol and Entisol, respectively (Table 8). In addition, there was an inversely proportional relationship between fruit firmness and size (R = −0.56) and a directly proportional relationship between fruit size and weight (R = 0.91) in the first week of harvest in the second season for ‘Legacy’. Fruit firmness in the second season on Andisol was 19% greater than on the other two soils (Table 8). Fruit size in the Inceptisol was 8.6% and 12.4% larger than in the Entisol and Andisol, respectively (Table 8). In addition, there was a directly proportional relationship between fruit size and weight (R = 0.99) in the second week of harvest in the second season for ‘Legacy’. All quality attributes decreased when comparing seasons; firmness was 3.4%, size was 3.8%, and weight was 4.4% lower compared to the first season (Table 8). Regarding the effect of increasing Ca rates on fruit size and fruit weight as an average of the different soils in each season (Table 9), there were significant differences only in the second season. However, the effects were erratic and could not explain the effects on these quality attributes.

Soil chemical properties evaluated at the end of the second year for ‘Duke’ were affected by soil type or location. Meanwhile, Ca rate only affected pH in fertility and saturated extract analyses and exchangeable Ca concentration (Table 10). The soil × Ca rate interaction affected the exchangeable Ca concentration. In the routine analyses, the highest pH was found in the Entisol (*p* < 0.05), followed by the Inceptisol, and the Andisol (*p* < 0.05) (Table 11). This ranking of values followed the same quantitative order as the initial soil analyses (Table 1). EC and exchangeable Ca concentrations were higher in the Inceptisol (*p* < 0.05) and lower in the Entisol (*p* < 0.05) (Table 11). The highest pH in the saturated extract analysis was also found in the Entisol (*p* < 0.05); there were no differences between the other two soils (Table 11). As in the routine analysis, the highest value of EC and Ca concentration in the saturated extract occurred in the Inceptisol (*p* < 0.05); however, the EC of the Entisol was similar to the value for the Inceptisol (Table 11). The highest values in the soil fertility and saturated soil extract analyses were consistent for the soil Ca concentrations evaluated (Table 11). Increasing Ca rates increased the pH in the fertility and saturated extract analyses and the exchangeable Ca concentration up to the 1 kg de Ca ha^−1^ rate (Table 12). The interaction soil × Ca rate showed effects only in the Inceptisol and Andisol where the exchangeable Ca concentration increased up to the 2 kg de Ca ha^−1^ rate (Table 13).

Soil chemical properties in the ‘Legacy’ trial were affected by the soil, except for EC determined in the saturated extract (Table 14). Ca rate only affected exchangeable concentration (fertility analysis) and available Ca (extract analysis) (Table 14). The highest pH in the fertility analysis occurred in Entisol (*p* < 0.05), followed by Andisol, which was higher than Inceptisol (*p* < 0.05) (Table 15). This ranking of values did not follow the same quantitative order as the initial soil analysis (Table 1). The highest EC was found in the Inceptisol, and the highest exchangeable Ca concentration was found in Andisol (Table 15). The ranking of exchangeable Ca concentration values was similar to the initial soil analysis (Table 1). The ranking of pH values for the saturated extract analysis followed the same order as for the fertility analysis and EC showed no differences between the soils (*p* > 0.05) (Table 15). Available Ca was higher in the Inceptisol (*p* < 0.05) and showed no differences with the other two soils. For the soil Ca concentrations evaluated, there was no effect between the highest values obtained in the soil fertility and saturated extract analyses (Table 15). The increasing dose of Ca increased the Ca concentration in the fertility and saturated extract analyses only at the dose of 4 kg de Ca ha^−1^ (Table 16).

## 4. Discussion

The chemical properties of the three soils were suitable for growing blueberries [21,22], except for the boron concentration in the Entisols, which was corrected by applying B in the fertigation program. The optimal Ca concentration in the three soils may explain the lack of a response in fruit quality attributes for both evaluated cultivars; only ‘Duke’ showed increased quantitative fruit weight in the second week of harvest. Although Angeletti et al. [23] reported a positive effect of soil Ca application on blueberry fruit quality attributes, the study did not provide any information regarding the concentration of soil Ca.

Values for both cultivars for fruit firmness, fruit size, and fruit weight were normal for the study area [5,7]; however, values were lower than the means for each cultivar reported by Strik et al. [3] at the North Willamette Research and Extension Center, Oregon State University, probably due to the different equipment used (Firm Pro respect de Firm Tech).

Differences in blueberry fruit quality trait values between locations and seasons have also been reported by other researchers [3,10,11,12,13]. They are usually associated with differences in orchard yield (higher yield results in smaller fruit size and lower fruit weight) and climatic differences between seasons (lower temperatures in spring negatively affect fruit size and weight, and higher temperatures during the fruit-filling phase lead to lower carbohydrate production, which negatively affects quality attributes). Considering that total yield and number of fruits per plant play an essential role in fruit size, weight, and firmness, and are often negatively correlated, changes in yield per plant may have a greater effect on fruit quality than calcium doses. However, the yield per plant in the cv. ‘Duke’ for the different rates of calcium was statistically similar, and in the first season corresponded to 3.3 ± 0.2 kg pl^−1^ in the Entisol, 3.6 ± 0.3 kg pl^−1^ in the Inceptisol, and 2.2 ± 0.2 kg pl^−1^ in the Andisol, while for the second season these values were 6.1 ± 0.2 kg pl^−1^ in the Entisol, 5.6 ± 0.2 kg pl^−1^ in the Inceptisol, and 6.5 ± 0.3 kg pl^−1^ in the Andisol [5]. For cv. ‘Legacy’, there was also no effect of the different calcium rates on yield per plant, and values corresponded to 5.0 ± 0.2 kg pl^−1^ in the Entisol, 3.8 ± 0.2 kg pl^−1^ in the Inceptisol, and 6.1 ± 0.3 kg pl^−1^ in the Andisol, while for the second season these values were 8.7 ± 0.2 kg pl^−1^ in the Entisol, 7.4 ± 0.1 kg pl^−1^ in the Inceptisol, and 4.6 ± 0.2 kg pl^−1^ in the Andisol [5]. Hancock et al. [41] indicated that lower air temperatures in spring and early summer may have affected blueberry production; in addition, an increase in air temperature between 20 and 25 °C increased CO_2_ assimilation. Therefore, temperatures below these values may reduce yield and affect fruit size and weight.

The present experiment showed a wider range of mean temperature and higher evaporation in the first season. However, fruit firmness in the first week of harvest for ‘Duke’ was similar between seasons, while fruit size was slightly lower and fruit weight was much lower in the first season. Both fruit firmness and fruit size were higher in the second week of harvest in the first season. For ‘Legacy’, fruit firmness in the first week of harvest was higher in the first season, while both fruit size and fruit weight were lower in the first season. At the second week of harvest, all quality attributes were higher in the first season. Therefore, the differences in quality attributes between seasons for both blueberry cultivars evaluated cannot be attributed to the higher mean temperature and evaporation recorded in the first season; this could be due to differences in yield, which was not evaluated in the present experiment.

The differences between the soils in the quality traits evaluated for both blueberry cultivars can be attributed to their different physical and chemical properties, as well as to the better overall condition for the blueberry crop in the Inceptisol and Andisol [21,22]. However, ‘Legacy’ showed higher fruit firmness in the first week of harvest in the first season in the Inceptisol and Andisol, which is explained by lower fruit weight. There is also an inverse relationship between fruit firmness and weight [2,3,5,12].

The inversely proportional relationships between fruit firmness and size and between fruit firmness and weight can be explained by the number and size of cells per fruit. This results in larger or smaller fruit size or weight, just as fruit firmness is mainly related to skin cell size and the shape of the underlying cell layers of the pericarp [42]. Therefore, a larger fruit could have larger cells and less skin consistency, reducing firmness. Larger fruit size and weight are associated with higher carbohydrate accumulation, which could be influenced by fruit load, as carbohydrate distribution to the fruit is greater with lower fruit load and lower yield [5,43]. The yields in the present experiment for the same soil, season, and blueberry cultivar were not affected by the calcium rate [5]. Redpath et al. [12] reported inversely proportional relationships between fruit firmness and size and between fruit firmness and weight for five blueberry cultivars. As fruit size increases, fruit weight is expected to increase because the increase in cell number or size during fruit growth also increases water and carbohydrate accumulation [14,44]. Several authors have noted the directly proportional relationship between fruit size and fruit weight in blueberry [8,12,14]. Differences in production between soils and seasons are affected by the previous season’s fruit load management and the potential or limitations of the soil for blueberry cultivation. Higher yields would have been expected on the Andisol soil compared to the other soils, but the Andisol used in this study has a shallow effective depth and is managed with a low ridge size, unlike the other two soils. This resulted in a lower average yield in the Andisol soil in both blueberry cultivars [5]. On the other hand, the Entisol soil has a greater effective depth and a larger ridge size, which resulted in a higher average yield in both blueberry cultivars [5].

The soil fertility analysis at the end of the experiment for ‘Duke’ showed differences in pH and exchangeable Ca concentrations among the soils, which followed the same ranking as in the initial analysis. However, the pH decreased in all three soils and exchangeable Ca concentration increased, except in the Andisol. The decrease in pH could be due to physiological reaction mechanisms generated by nutrient extraction during two seasons (excretion of H^+^ from the roots to compensate for the charge gain by cation uptake), the acidifying effect of carboxylic acids that are part of the applied calcium fertilizer, and the excretion effect of organic compounds from the roots [35,45,46]. The increase in exchangeable Ca concentration is partly in response to the applied Ca fertilizer and cation exchange processes due to the consumption of N as ammonia by the blueberry crop [47,48,49]. The EC showed differences between soils, which were mainly related to the physicochemical properties of each soil. These properties produce differences in nutrient adsorption and desorption capacity; a lower adsorption capacity in Entisol and a higher risk of nutrient leaching were due to their textural composition [46]. The saturated extract analyses showed a difference in pH between the soils, which followed the same trend as the fertility analysis. However, the EC in the extract had higher values than those of the fertility analysis due to the technique used to obtain the extract to perform the analysis (water-soluble ions desorbed after saturating the soil, which were washed out of the saturated matrix). The available Ca from the extract showed lower values compared to the fertility analysis because the Ca desorption in a distilled water solution is very low and lower than that of the other cations [46]. There was a relationship between the highest exchangeable and available Ca values in the soils, except in the Entisol, which had the same available Ca concentration as the Andisol. This could be explained by the higher cation desorption of the Entisol due to its lower buffering capacity [46]. Increasing Ca rates applied to the soil had a directly proportional effect on exchangeable Ca (R = 0.82) and increased pH in fertility and saturated extract analyses. However, a decrease in pH would have been expected in both types of analyses due to the acidifying effect produced by the application of carboxylic acids. The increase in pH in the fertility and saturated extract analyses could be partially explained by the increase in soil Ca concentration and its effect on changes in H^+^ and OH^−^ concentrations [45,46]. Finally, the three soils responded differently to changes in exchangeable Ca concentration with increasing Ca rates; which is explained by soil physicochemical properties and their effect on cation adsorption and desorption capacity [45,46].

As for the experiment with ‘Legacy’, soil pH in the fertility and saturated extract analyses showed the same ranking among soils. However, the pH ranking among soils in the fertility analysis differed from the pH in the initial analysis for the Inceptisol and Andisol. This could be explained by yield differences in each soil, as higher yields result in higher nutrient extraction and higher H^+^ excretion, which decreases soil pH [35,45,46]. However, the yields in the present experiment for the same soil, season and blueberry cultivar were not affected by the calcium rate [5]. EC differences between soils for fertility analysis were similar to those mentioned above in the ‘Duke’ experiment. Exchangeable Ca in ‘Legacy’ did not show the same behavior as in ‘Duke’; values followed the same trend of ranking by concentration as in the initial soil analysis. As expected, soil pH in the saturated extract analysis followed the same trend as in the fertility analysis. Differences between soils would have been expected for EC associated with their physicochemical properties [46], but this did not occur. Available Ca concentration did not show the same range of values behavior observed for the exchangeable Ca analysis; available Ca was higher in the Inceptisol, which could respond to a higher Ca adsorption and desorption capacity in this soil associated with its physicochemical properties [45,46]. Finally, increasing the Ca rate applied to the soil had a directly proportional effect on both exchangeable Ca (R = 0.84) and available Ca (R = 0.89), which was expected due to the Ca binding capacity in the cation exchange capacity of soils [45,46].

## 5. Conclusions

Under the conditions of the present study, the application of increasing calcium (Ca) rates in the selected range (0.5 to 4 kg ha^−1^ as a carboxylic acid formulation) did not affect the quality attributes of firmness, size, and weight of ‘Duke’ and ‘Legacy’ blueberry fruits. Fruit firmness, size, and weight showed differences between seasons and between locations or soil types. The highest values for ‘Duke’ were firmness between 164 and 186 g mm^−1^, size between 15.7 and 16.9 mm, and fruit weight between 1.60 and 1.76 g, whereas ‘Legacy’ showed firmness values between 163 and 173 g mm^−1^, fruit size between 16.2 and 17.2 mm, and fruit weight between 2.01 and 2.40 g. Soil Ca application at low rates increased exchangeable Ca concentration in both blueberry cultivars, partially increased soil pH in ‘Duke’ and increased the soil available Ca concentration in ‘Legacy’.

## Figures and Tables

**Table 1 plants-13-00001-t001:** Soil chemical properties at 0–30 cm depth prior to the start of the experiment (2020 season) in three soils.

Parameters	Soil		
	Entisol	Inceptisol	Andisol
Location	Santa Cruz de Cuca	Larqui	Capilla
Clay (%)	3.4	21.1	20.0
Silt (%)	4.2	47.8	21.4
Sand (%)	92.4	31.0	58.6
pH _(soil:water 1:2.5)_	6.24	5.51	5.35
Organic matter (g kg^−1^)	1.21	7.73	7.86
Available N (mg kg^−1^)	9.1	18.2	19.8
Olsen P (mg kg^−1^)	40.5	95.4	62.3
Exchangeable K (cmol_+_ kg^−1^)	0.39	0.77	0.78
Exchangeable Ca (cmol_+_ kg^−1^)	2.49	5.86	6.08
Exchangeable Mg (cmol_+_ kg^−1^)	1.15	1.15	1.25
Exchangeable Na (cmol_+_ kg^−1^)	0.20	0.32	0.19
Exchangeable Al (cmol_+_ kg^−1^)	0.01	0.08	0.06
Available S (mg kg^−1^)	26.4	223.4	56.9
Available Fe (mg kg^−1^)	26.2	51.6	48.9
Available Mn (mg kg^−1^)	2.1	6.8	4.8
Available Zn (mg kg^−1^)	15.5	12.2	28.9
Available Cu (mg kg^−1^)	3.8	1.7	3.4
Available B (mg kg^−1^)	0.11	1.44	0.49

N: nitrogen; P: phosphorus; K: potassium; Ca: calcium; Mg: magnesium; Na: sodium; Al: aluminum; S: sulfur; Fe: iron; Mn: manganese; Zn: zinc; Cu: copper; B: boron.

**Table 2 plants-13-00001-t002:** Significance tests for quality attributes at the first and second weeks of harvest and fruit yield for ‘Duke’ as affected by different seasons, soils, and calcium rates.

Source of Variation	First Week of Harvest			Second Week of Harvest		
	Firmness	Size	Weight	Firmness	Size	Weight
Season (Y)	NS	**	**	**	**	**
Soil (S)	**	**	**	**	**	**
Calcium rate (Ca)	NS	NS	NS	NS	*	**
Y × S	**	**	**	**	*	NS
Y × Ca	NS	NS	NS	NS	NS	NS
S × Ca	NS	NS	NS	NS	NS	NS
Y × S × Ca	NS	NS	NS	NS	NS	NS

* Significant at *p* < 0.05; ** Significant at *p* < 0.01. NS: Nonsignificant.

**Table 3 plants-13-00001-t003:** Effect of season × soil interaction on fruit quality attributes in the first week of harvest for ‘Duke’ as a mean of different calcium rates.

Season	Soil	Firmness (g mm^−1^)	Size (mm)	Weight (g)
1	Entisol	161 ± 1.6 b	14.9 ± 0.09 b	1.40 ± 0.02 b
	Inceptisol	183 ± 2.5 a	15.8 ± 0.09 a	1.60 ± 0.02 a
	Andisol	154 ± 1.5 b	14.5 ± 0.10 c	1.33 ± 0.02 b
2	Entisol	158 ± 1.6 c	14.4 ± 0.12 b	1.42 ± 0.03 b
	Inceptisol	172 ± 2.2 a	15.7 ± 0.15 a	1.70 ± 0.04 a
	Andisol	165 ± 1.4 b	15.8 ± 0.08 a	1.76 ± 0.02 a

Different letters in the same column for the same season indicate statistical differences between soils according to Tukey’s test (*p* < 0.05). Values are means ± standard error. n = 75 for each season.

**Table 4 plants-13-00001-t004:** Effect of season × soil interaction on fruit firmness and size in the second week of harvest for ‘Duke’ as a mean of different calcium rates.

Season	Soil	Firmness (g mm^−1^)	Size (mm)	Weight (g)
1	Entisol	140 ± 2.0 c	14.1 ± 0.10 c	1.35 ± 0.03 b
	Inceptisol	186 ± 1.6 a	16.9 ± 0.09 a	1.90 ± 0.03 a
	Andisol	161 ± 1.5 b	16.1 ± 0.10 b	1.80 ± 0.03 a
2	Entisol	143 ± 1.6 c	13.7 ± 0.14 b	1.25 ± 0.03 b
	Inceptisol	164 ± 2.7 a	15.9 ± 0.22 a	1.81 ± 0.06 a
	Andisol	150 ± 1.6 b	15.6 ± 0.13 a	1.74 ± 0.04 a

Different letters in the same column for the same season indicate statistical differences between soils according to Tukey’s test (*p* < 0.05). Values are means ± standard error. n = 75 for each season.

**Table 5 plants-13-00001-t005:** Effect of the season, soil, and calcium rate on fruit weight in the second week of harvest for ‘Duke’.

Source of Variation	Comparisons for Each Source of Variation	Fruit Weight (g)
Season	1	1.68 ± 0.03 a
	2	1.60 ± 0.04 b
Soil	Entisol	1.30 ± 0.04 b
	Inceptisol	1.85 ± 0.03 a
	Andisol	1.77 ± 0.03 a
Calcium rate (kg ha^−1^)	0	1.53 ± 0.05 b
	0.5	1.64 ± 0.06 ab
	1.0	1.68 ± 0.05 a
	2.0	1.66 ± 0.06 ab
	4.0	1.69 ± 0.06 a

Different letters in the same column indicate statistical differences between seasons, soils, or calcium rates according to Tukey’s test (*p* < 0.05). Values are means ± standard error. n = 75 for each season, 50 for each soil, and 30 for each Calcium rate.

**Table 6 plants-13-00001-t006:** Significance testing for quality attributes in the first and second weeks of harvest and fruit yield for ‘Legacy’ as affected by different seasons, soils, and calcium rates.

Source of Variation	First Week of Harvest			Second Week of Harvest		
	Firmness	Size	Weight	Firmness	Size	Weight
Season (Y)	**	**	**	**	**	**
Soil (S)	**	**	**	**	**	**
Calcium rate (Ca)	NS	NS	NS	NS	NS	NS
Y × S	**	**	**	**	**	**
Y × Ca	NS	NS	NS	NS	**	*
S × Ca	NS	NS	NS	NS	NS	NS
Y × S × Ca	NS	NS	NS	NS	NS	NS

* Significant at *p* < 0.05; ** Significant at *p* < 0.01. NS: Nonsignificant.

**Table 7 plants-13-00001-t007:** Effect of season × soil interaction on fruit quality attributes in the first week of harvest for ‘Legacy’ as a mean of different calcium rates.

Season	Soil	Firmness (g mm^−1^)	Size (mm)	Weight (g)
1	Entisol	166 ± 1.7 a	15.8 ± 0.11 b	1.86 ± 0.04 ab
	Inceptisol	150 ± 1.6 b	16.2 ± 0.08 a	2.01 ± 0.02 a
	Andisol	172 ± 2.9 a	15.8 ± 0.14 b	1.77 ± 0.06 b
2	Entisol	167 ± 1.8 a	16.2 ± 0.10 b	2.05 ± 0.04 b
	Inceptisol	139 ± 1.6 c	17.2 ± 0.15 a	2.40 ± 0.05 a
	Andisol	159 ± 2.6 b	15.7 ± 0.17 b	1.86 ± 0.05 c

Different letters in the same column for the same season indicate statistical differences between soils according to Tukey’s test (*p* < 0.05). Values are means ± standard error. n = 75 for each season.

**Table 8 plants-13-00001-t008:** Effect of season × soil interaction on fruit quality attributes in the second week of harvest for ‘Legacy’ as a mean of different calcium rates.

Season	Soil	Firmness (g mm^−1^)	Size (mm)	Weight (g)
1	Entisol	163 ± 1.3 a	15.1 ± 0.10 b	1.53 ± 0.03 c
	Inceptisol	149 ± 1.3 b	17.0 ± 0.08 a	2.17 ± 0.03 a
	Andisol	168 ± 2.1 a	15.4 ± 0.09 b	1.79 ± 0.03 b
2	Entisol	145 ± 1.6 b	14.9 ± 0.10 b	1.68 ± 0.03 b
	Inceptisol	145 ± 1.0 b	16.3 ± 0.12 a	2.08 ± 0.04 a
	Andisol	173 ± 6.3 a	14.5 ± 0.08 c	1.49 ± 0.03 c

Different letters in the same column for the same season indicate statistical differences between soils according to Tukey’s test (*p* < 0.05). Values are means ± standard error. n = 75 for each season.

**Table 9 plants-13-00001-t009:** Effect of season × calcium rate interaction on fruit size and weight in the second week of harvest for ‘Legacy’ as a mean of different soils.

Season	Calcium Rate (kg ha^−1^)	Size (mm)	Weight (g)
1	0	15.8 ± 0.7 a	1.82 ± 0.19 a
	0.5	15.9 ± 0.5 a	1.84 ± 0.16 a
	1.0	15.7 ± 0.5 a	1.81 ± 0.16 a
	2.0	15.9 ± 0.5 a	1.85 ± 0.20 a
	4.0	15.8 ± 0.7 a	1.82 ± 0.23 a
2	0	15.3 ± 0.4 ab	1.74 ± 0.14 ab
	0.5	15.4 ± 0.6 a	1.80 ± 0.20 a
	1.0	15.4 ± 0.7 a	1.80 ± 0.21 a
	2.0	14.9 ± 0.5 b	1.63 ± 0.15 b
	4.0	15.3 ± 0.6 ab	1.76 ± 0.16 ab

Different letters in the same column for the same season indicate statistical differences between calcium rates according to Tukey’s test (*p* < 0.05). Values are means ± standard error. n = 75 for each season and 15 for each Calcium rate.

**Table 10 plants-13-00001-t010:** Significance testing for soil properties at the end of the second year for ‘Duke’ as affected by different soils and calcium rates.

Source of Variation	Soil Fertility Analysis			Saturated Soil Extract Analysis		
	pH	EC	Exchangeable Ca	pH	EC	Available Ca
Soil (S)	**	**	**	**	*	**
Calcium rate (Ca)	**	NS	**	**	NS	NS
S × Ca	NS	NS	**	NS	NS	NS

* Significant at *p* < 0.05; ** Significant at *p* < 0.01. NS: Nonsignificant; EC: electrical conductivity.

**Table 11 plants-13-00001-t011:** Effect of soil-on-soil properties at the end of the second year for ‘Duke’ as a mean of five calcium (Ca) rates.

Soil Property		Entisol	Inceptisol	Andisol
Soil fertility analysis	pH _(soil:water 1:2.5)_	6.89 ± 0.06 a	5.93 ± 0.06 b	5.67 ± 0.03 c
	EC (dS m^−1^)	0.02 ± 0.002 c	0.09 ± 0.006 a	0.04 ± 0.002 b
	Exchangeable Ca (cmol_+_ kg^−1^)	2.89 ± 0.10 c	9.64 ± 0.47 a	4.81 ± 0.21 b
Saturated soil extract analysis	pH _(soil:water 1:5)_	6.99 ± 0.08 a	5.47 ± 0.10 b	5.29 ± 0.08 b
	EC (dS m^−1^)	0.26 ± 0.03 ab	0.31 ± 0.02 a	0.23 ± 0.01 b
	Available Ca (mg L^−1^)	1.01 ± 0.07 b	1.60 ± 0.12 a	1.01 ± 0.05 b

Different letters in the same file indicate statistical differences between soils according to Tukey’s test (*p* < 0.05). EC: Electrical conductivity. Values are means ± standard error. n = 25 for each soil.

**Table 12 plants-13-00001-t012:** Effect of the calcium rate on soil properties at the end of the second year for ‘Duke’ as a mean of three soils.

Soil Property		Ca Rate (kg ha^−1^)				
		0	0.5	1.0	2.0	4.0
Soil fertility analysis	pH _(soil:water 1:2.5)_	5.96 ± 0.14 b	6.17 ± 0.16 ab	6.24 ± 0.16 a	6.23 ± 0.14 a	6.21 ± 0.15 a
	Exchangeable Ca (cmol_+_ kg^−1^)	4.53 ± 0.66 b	5.45 ± 0.69 ab	6.02 ± 0.92 a	6.39 ± 0.94 a	6.50 ± 0.91 a
Saturated soil extract analysis	pH _(soil:water 1:5)_	5.45 ± 0.20 b	5.81 ± 0.23 ab	6.04 ± 0.23 a	6.15 ± 0.21 a	6.13 ± 0.21 a

Different letters in the same row indicate statistical differences between calcium rates according to Tukey’s test (*p* < 0.05). Values are means ± standard error. n = 15 for each Ca rate.

**Table 13 plants-13-00001-t013:** Effect of the season × calcium rate interaction on soil exchangeable Ca at the end of the second year for ‘Duke’.

Ca Rate (kg ha^−1^)	Soil		
	Entisol	Inceptisol	Andisol
0	2.60 ± 0.31 a	7.23 ± 1.05 c	3.77 ± 0.43 b
0.5	2.83 ± 0.20 a	8.62 ± 0.48 bc	4.91 ± 0.69 ab
1.0	2.89 ± 0.13 a	10.10 ± 1.44 ab	5.06 ± 0.31 a
2.0	2.94 ± 0.02 a	11.16 ± 0.42 a	5.07 ± 0.43 a
4.0	3.20 ± 0.29 a	11.09 ± 0.53 a	5.22 ± 0.28 a

Different letters in the same column indicate statistical differences between calcium (Ca) rates according to Tukey’s test (*p* < 0.05). Values are means ± standard error. n = 5 for each soil and Ca rate.

**Table 14 plants-13-00001-t014:** Significance testing for soil properties at the end of the second year for ‘Legacy’ as affected by different soils and calcium rates.

Source of Variation	Soil Fertility Analysis			Saturated Soil Extract Analysis		
	pH	EC	Exchangeable Ca	pH	EC	Available Ca
Soil (S)	**	**	**	**	NS	**
Calcium rate (Ca)	NS	NS	*	NS	NS	**
S × Ca	NS	NS	NS	NS	NS	NS

* Significant at *p* < 0.05; ** Significant at *p* < 0.01. NS: Nonsignificant; EC: electrical conductivity.

**Table 15 plants-13-00001-t015:** Effect of soil-on-soil properties at the end of the second year for ‘Legacy’ as a mean of five calcium (Ca) rates.

Soil Property		Entisol	Inceptisol	Andisol
Soil fertility analysis	pH _(soil:water 1:2.5)_	6.95 ± 0.07 a	5.75 ± 0.05 c	6.06 ± 0.04 b
	EC (dS m^−1^)	0.03 ± 0.002 c	0.08 ± 0.005 a	0.05 ± 0.003 b
	Exchangeable Ca (cmol_+_ kg^−1^)	2.83 ± 0.11 c	8.52 ± 0.22 b	9.40 ± 0.27 a
Saturated soil extract analysis	pH _(soil:water 1:5)_	7.10 ± 0.06 a	5.53 ± 0.06 c	6.06 ± 0.09 b
	EC (dS m^−1^)	0.22 ± 0.03 a	0.24 ± 0.02 a	0.25 ± 0.02 a
	Available Ca (mg L^−1^)	0.93 ± 0.11 b	1.29 ± 0.10 a	1.07 ± 0.03 b

Different letters in the same row indicate statistical differences between soils according to Tukey’s test (*p* < 0.05). EC: Electrical conductivity. Values are means ± standard error. n = 25 for each soil.

**Table 16 plants-13-00001-t016:** Effect of the calcium (Ca) rate on soil properties at the end of the second year for ‘Legacy’ as a mean of three soils.

Soil Property		Ca Rate (kg ha^−1^)				
		0	0.5	1.0	2.0	4.0
Soil fertility analysis	Exchangeable Ca (cmol_+_ kg^−1^)	6.48 ± 0.77 b	6.91 ± 0.84 ab	6.92 ± 0.81 ab	7.16 ± 0.79 ab	7.39 ± 0.82 a
Saturated soil extract analysis	Available Ca (mg L^−1^)	0.88 ± 0.12 b	1.09 ± 0.11 ab	1.09 ± 0.12 ab	1.14 ± 0.09 ab	1.28 ± 0.12 a

Different letters in the same row indicate statistical differences between calcium rates according to Tukey’s test (*p* < 0.05). Values are means ± standard error. n = 15 for each Ca rate.

## Data Availability

Data are contained within the article.

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
