# Peer review of "Can the Firmness, Weight, and Size of Blueberry Fruit Be Enhanced through the Application of Low Amounts of Calcium to the Soil?"

_plants, 2023, doi:10.3390/plants13010001_

Round 1

Reviewer 1 Report

Comments and Suggestions for Authors

All the comments are given in the manuscript.

Author Response

Line 25: keywords. The Word blueberry was deleted.

Line 69: The scientific name was write in italic.

Line 100: the word blueberry was included.

Line 120: the value correct is 1:2.5, which was corrected in all the text and the Tables.

Line 120: the word organic carbon was delete and changed by the word organic matter (OM), similar to the Table 1.

Line 121: the word “inorganic” was changed by the word “available” similar to the Table 1.

Line 172: the value 1:2.5 was corrected by 1:5.

Table 11. The relationship soil:water for the soil fertility analysis was reviewed and the correct value is 1:2.5.

Table 12. The relationship soil:water for the soil fertility analysis was reviewed and the correct value is 1:2.5.

Reviewer 2 Report

Comments and Suggestions for Authors

The paper presents thorough analysis of Calcium effect on fruit weight, size and firmness. Quite large data is presented and well discussed. However, the total yield and fruit number per plant play an essential role for all fruit quality characteristics, and usually corelates negatively with fruit size and weight. It should be quite large variation of total yield, since the trial was conducted during two seasons at three different sites and two cultivars. So, without yield evaluation, interpretations of Calcium effects on fruit quality are meaningless.

Author Response

For answer to the advice and correction of the reviewer number 2 was included the next paragraph in the Discussion:

Lines 306-316:

Considering that total yield and number of fruits per plant play an essential role in fruit size, weight and firmness, and are often negatively correlated, changes in yield per plant may have a greater effect on fruit quality than calcium doses. However, the yield per plant in the cv. Ê»Dukeʼ for the different rates of calcium was statistically similar, and in the first season corresponded to 3.3 ± 0.2 kg pl-1 in the Entisol, 3.6 ± 0.3 kg pl-1 in the Inceptisol, and 2.2 ± 0.2 kg pl-1 in the Andisol, while for the second season these values were 6.1 ± 0.2 kg pl-1 in the Entisol, 5.6 ± 0.2 kg pl-1 in the Inceptisol, and 6.5 ± 0.3 kg pl-1 in the Andisol [5]. For cv. Ê»Legacyʼ there was also no effect of the different calcium rates on yield per plant, and values corresponded to 5.0 ± 0.2 kg pl-1 in the Entisol, 3.8 ± 0.2 kg pl-1 in the Inceptisol, and 6.1 ± 0.3 kg pl-1 in the Andisol, while for the second season these values were 8.7 ± 0.2 kg pl-1 in the Entisol, 7.4 ± 0.1 kg pl-1 in the Inceptisol, and 4.6 ± 0.2 kg pl-1 in the Andisol [5].

Round 2

Reviewer 2 Report

Comments and Suggestions for Authors

Authors provided requested information on the yield, but more thorough discussions on the relationship between yield, and soil and fruit quality parameters are needed.

Please, provide arguments on your hypothesis that low doses of soil applied Calcium can have (but did not) effect on fruit quality. With the same success it was possible to test all known minerals and declare that they have no effect. So, what is the main message of this manuscript? It seems it is not Calcium? Consider, to make emphasis on different soil properties and cultivar behaviour, but not on Calcium.

Introduction too long. Too many numbers and presented results. Such presentations better suits to section Discussions.

Lines 96 97. What is a period ‘’between flowering and fruit with little growth’’? Describe in BBCH stages.

Materials.

Line 137. What means the yield measurement in Mg ha-1 ?

Line 150. ‘’fruit of 5 cm diameter’’ – unbelievable large fruits.

Lines 157 159. Strange to read ‘’The remaining 20% of the ‘Legacy’ fruit was mechanically harvested due to inferior quality during the third week of harvest’’, knowing that experiment was conducted in two years and at three different sites. Does it common harvest practice of Legacy variety?

Table 4 add results of fruit weight.

Authors provided requested information on the yield, but more thorough discussions on the relationship between yield, and soil and fruit quality parameters are needed.

Author Response

Dear Reviewer 2, I thank you very much for your suggestions and corrections, and the answers and corrections made are highlighted in grey in the text and are as follows:

1. Please, provide arguments on your hypothesis that low doses of soil applied Calcium can have (but did not) effect on fruit quality. With the same success it was possible to test all known minerals and declare that they have no effect. So, what is the main message of this manuscript? It seems it is not Calcium? Consider, to make emphasis on different soil properties and cultivar behaviour, but not on Calcium.

Line 95-102 was inclueded the follow paragraph:

Considering that the first phase of fruit growth is characterized by rapid cell division [34], and that this process is promoted by the presence of calcium and cytokinins [35] , the application of low doses of calcium with sources that facilitate its availability in early stages of flower and fruit development, associated with low natural availability when soil temperature is low, could contribute to improve fruit firmness, the hypothesis of our work is that the application of Ca to the soil at lower rates through the formulation of carboxylic acids between flowering and 14 days after flowering can improve some quality attributes in fruit and also affect some chemical properties of the soil.

2. Introduction too long. Too many numbers and presented results. Such presentations better suits to section Discussions.

I agree that in the introduction there is a lot of information that is then cited again as coinciding with or differing from the results in the discussion (without repeating the information), however I consider it necessary to provide details of the results obtained by each investigation in order to contribute to the contribution of condensed information on this subject, given that there is little literature that refers to similar work with calcium.

3. Lines 96 97. What is a period ‘’between flowering and fruit with little growth’’? Describe in BBCH stages.

The correction include was the next:

Line 114: between flowering and 14 days after flowering .

Line 153-155: The treatments evaluated were five doses of Ca fertilization applied to the soil and dissolved in water (simulating fertigation application); the doses were 0, 0.5, 1.0, 2.0, and 4.0 kg Ca ha-1, which were totally applied between start of flowering and 14 days after flowering.

4. Materials.

4.1. Line 137. What means the yield measurement in Mg ha-1 ?

Mg ha-1 means Ton ha-1. The unit of measure could be changed if it is neccesary.

4.2. Line 150. ‘’fruit of 5 cm diameter’’ – unbelievable large fruits. 

To avoid confusion the sentence was changed and the new sentence is "between start of flowering and 14 days after flowering".

4.3. Lines 157 159. Strange to read ‘’The remaining 20% of the ‘Legacy’ fruit was mechanically harvested due to inferior quality during the third week of harvest’’, knowing that experiment was conducted in two years and at three different sites. Does it common harvest practice of Legacy variety?.

It is indeed a common practice in the Legacy variety given the drop in quality of the fruit in the last harvesting period.

4.4. Table 4 add results of fruit weight.

The results were added.

Table 4. Effect of season × soil interaction on fruit firmness and size in the second week of harvest for Ê»Dukeʼ as a mean of different calcium rates.

Season

Soil

Firmness (g mm-1)

Size (mm)

Weight (g)

1

Entisol

140±2.0 c

14.1±0.10 c

1.35±0.03 b

Inceptisol

186±1.6 a

16.9±0.09 a

1.90±0.03 a

Andisol

161±1.5 b

16.1±0.10 b

1.80±0.03 a

2

Entisol

143±1.6 c

13.7±0.14 b

1.25±0.03 b

Inceptisol

164±2.7 a

15.9±0.22 a

1.81±0.06 a

Andisol

150±1.6 b

15.6±0.13 a

1.74±0.04 a

Different letters in the same column for the same season indicate statistical differences between soils according to Tukey’s test (p < 0.05). Values are means ± standard error.

n = 75 for each season.

5. Authors provided requested information on the yield, but more thorough discussions on the relationship between yield, and soil and fruit quality parameters are needed.

The paragraph included was the follow:

Line 254-260: 

Differences in production between soils and seasons are affected by the previous season's fruit load management and the potential or limitations of the soil for blueberry cultivation. Higher yields would have been expected on the Andisol soil compared to the other soils, but the Andisol used in this study has a shallow effective depth and is managed with a low ridge size, unlike the other two soils. This resulted in a lower average yield in the Andisol soil in both blueberry cultivars [5]. On the other hand, the Entisol soil has a greater effective depth and a larger ridge size, which resulted in a higher average yield in both blueberry cultivars [5].

In addition, between lines 341 to 354, the relationship between yield and fruit quality attributes is discussed, referring to the results found between these parameters, generally with inversely proportional relationships, and directely proportional relationships between fruit size and weight.

Round 3

Reviewer 2 Report

Comments and Suggestions for Authors

 Mg ha-1  means megagram per hectare. Yield per hectare can not be measured in grams. use t ha-1 or Mt ha-1

Author Response

Dear Reviewer number 2, the suggestion "Mg ha-1  means megagram per hectare. Yield per hectare can not be measured in grams. use t ha-1 or Mt ha-1" was considered and the new sentence is presented in the Lines 141-143 and highlighted in blue color.

Line 141-143: The age of the orchard ranged from 6 to 10 years (orchards at the peak of production with yields per plant between 10 and 15 t ha-1 for ‘Duke’ and 15 and 20 t ha-1 for ‘Legacy’). 
